# Gut Microbiota and Mucosal Immunity in the Neonate

**DOI:** 10.3390/medsci6030056

**Published:** 2018-07-17

**Authors:** Majda Dzidic, Alba Boix-Amorós, Marta Selma-Royo, Alex Mira, Maria Carmen Collado

**Affiliations:** 1Department of Biotechnology, Institute of Agrochemistry and Food Technology-Spanish National Research Council (IATA-CSIC), 46980 Valencia, Spain; majda.dzidic@iata.csic.es (M.D.); albaboix@iata.csic.es (A.B.-A.); mselma@iata.csic.es (M.S.-R.); 2Department of Health and Genomics. Center for Advanced Research in Public Health, FISABIO Foundation, 46020 Valencia, Spain

**Keywords:** gut microbiota, postnatal immune development, gut immunity, breastfeeding, probiotics, antibiotics

## Abstract

Gut microbiota colonization is a complex, dynamic, and step-wise process that is in constant development during the first years of life. This microbial settlement occurs in parallel with the maturation of the immune system, and alterations during this period, due to environmental and host factors, are considered to be potential determinants of health-outcomes later in life. Given that host–microbe interactions are mediated by the immune system response, it is important to understand the close relationship between immunity and the microbiota during birth, lactation, and early infancy. This work summarizes the evidence to date on early gut microbiota colonization, and how it influences the maturation of the infant immune system and health during the first 1000 days of life. This review will also address the influence of perinatal antibiotic intake and the importance of delivery mode and breastfeeding for an appropriate development of gut immunity.

## 1. Introduction

Epidemiological studies highlight the relevance of the period from conception to early life in the physiological and structural patterns of infant development, affecting their potential “health programming”. The fetus adapts to the intrauterine environment, being able to alter its metabolism in response to external stimuli. The physiological and metabolic adaptations that the fetus undergoes in response to those stimuli could produce permanent changes in the host, which may lead to a higher risk of developing diseases and/or disorders, such as obesity, allergies, diabetes, or cardiovascular diseases, in adult life [1].

The first 1000 days after conception (including the pregnancy period and the first two years of life), which are considered a “window of opportunity”, are crucial for the development and health of the future adult, as well as key to the establishment of the intestinal microbiota and immune system maturation. The physiological and immune development of the infant and the establishment of their microbiota occur in parallel throughout this short space of time. This microbiota plays a central role in health, intervening in key host metabolic and immunological functions.
DEFINITIONS**Microbiota:** the microbial community in a specific niche/environment.**Microbiome**: the total genomic repertoire of a microbial community (microbiota).

## 2. Prenatal Microbial Exposure and Mother–Fetus Immune Interaction

Recent studies have shown that the physiologic, immune, and metabolic changes that occur during pregnancy run in parallel with variations in microbial composition and diversity in the maternal microbiota [2,3,4]. During gestation, the maternal immune system adapts to the required tolerance between mother and fetus, although the microbiota–immune system interaction during pregnancy has been poorly described. Moreover, maternal conditions, including pre-gestational body mass index [5], weight gain during pregnancy [6], various diseases, such as gestational diabetes [7] or allergy [8,9], and especially antibiotic consumption [10] may affect the microbiota composition of the mother and newborn. However, the relevance of the shifts that these conditions may provoke in the immune system development is still unclear. Despite the importance of diet on the adult the microbiota, only a few studies [11,12,13] have investigated how the microbiota of pregnant women affect newborn colonization or its relation to immune system maturation.

It has been suggested that the maternal microbiota and its metabolites transferred to the fetus may play a key role in a newborn’s preparation for optimal host–microbe interactions, thus influencing infant immune responses [14]. In this scenario, the maternal gestational environment is critical for fetal physiology and development, inducing long-lasting and/or permanent modifications that may imprint a specific hallmark on the fetus, leading to an increased risk of developing non-communicable diseases later in life [1].

It has been suggested that that microbial contact may start before birth, although this hypothesis is still controversial. The original idea that the fetus resides in a sterile environment has being challenged [15], primarily by the presence of viable bacteria in the meconium [16,17,18,19], thus reflecting a potential in utero microbial environment, and also by the presence of bacteria and bacterial DNA in the maternal–fetal interphase. A few studies have examined the potential bacterial transmission through the placental barrier in healthy term pregnancies, analyzing umbilical cord [20], fetal membranes [21,22], and amniotic fluid [21,23] from healthy neonates. Indeed, a strong correlation between intrauterine infections and preterm deliveries has been reported, and remarkably, most of the bacteria detected in these infections are common habitants of the vaginal tract [24,25]. Nevertheless, it has to be kept in mind that the bacterial load in the fetal environment has been estimated to be extremely low, facilitating the possibility for (environmental) contamination of the studied samples. Thus, some authors consider that the placenta and amniotic fluid are sterile [26,27].

The bacteria that have been detected in meconium include *Enterococcus* and *Escherichia* and partly resemble the maternal and infant gastrointestinal tract microbiota [16,19]. Maternal microbial transmission has been explored in an animal study where genetically-labelled *Enterococcus faecium*, administered to pregnant mice were found in the meconium [16]. However, the mechanisms by which gut bacteria enter the uterine environment need further investigation, especially in humans, where both technical and ethical difficulties are encountered. Nevertheless, it has been speculated that the bacteria might access the bloodstream and travel to the placenta through translocation of the gut epithelium, presumably accompanied by dendritic cells, as shown in mice models [28,29]. Some mechanisms have been proposed for the microbiota spreading, including migration from the vagina or hematogenous spread from the gut and oral microbiota [30]. In human studies, the placental microbiota has been described to resemble the human oral microbiota more than the vaginal, fecal, skin, or nasal microbiota, hosting non-pathogenic commensal microbiota from the Firmicutes, Tenericutes, Proteobacteria, Bacteroidetes, and Fusobacteria phyla [31]. Speculatively, the microbiota here is established via the hematogenous spread of oral microbiota, likely during early vascularization and placentation. Moreover, oral cavity bacteria have also been isolated from the amniotic fluid where they could enter via the bloodstream during periodontal infections, which increase in frequency during pregnancy [23,32,33]. It has also been proposed that oral bacteria can reach the uterine and vaginal environment through the bloodstream and may induce and/or influence the labor process. In a study by Dasanayake et al., it was observed that specific maternal oral bacteria, namely *Actinomyces naeslundii*, was associated with prematurity and lower birth weight, while others, such as lactobacilli, were linked with term deliveries and higher birth weight [34]. Indeed, the microbiota could have an impact on the risk for preterm delivery, as periodontal treatment during pregnancy reduced the rate of premature birth from 10% to 1.8% [35]. Many studies have focused on the role of the maternal vaginal [36] and oral microbiota [37], or even the placental microbiota, in preterm delivery [38,39]. However, these studies do not adress the possible immune mechanisms involved in premature delivery, which could be mediated by pro-inflammatory molecules from a microbial source or by direct exposure to microbial antigens due to bacterial translocation from bleeding gingiva, triggering an autoimmune response that could lead to a premature delivery. Nevertheless, it is still under debate whether the microbes could really reach the gestational cavity and, if so, whether they may have any beneficial effects for pregnancy outcomes and future health status for neonates. Indeed, most of these studies have been performed using culture-independent techniques of which the results are commonly affected by contamination. However, a few studies have isolated cultivable obligated anaerobes from meconium samples [17], which are unlikely to be a contamination during delivery. Furthermore, metabolically active bacteria in the meconium have also been shown by RNA sequencing analysis [40]. Despite these recent results, more studies about the presence of bacteria in the meconium, placenta and amniotic fluid are needed in order to confirm the hypothesis of the in utero colonization.

## 3. Perinatal Factors Influencing Infant Microbiota Development and Immune System Maturation

Neonatal colonization is a fragile, dynamic, and step-wise process and may be affected by several maternal and neonatal factors (Figure 1). It has been reported that the effect of perinatal factors, including the mode of delivery, breastfeeding, and antibiotics use early in life, could influence the microbiota composition during a short period of time [41,42,43,44,45,46]. For instance, *Lactobacillus*, *Bifidobacterium*, *Bacteroides*, *Clostridium*, and *Streptococcus*, which are commonly influenced by the above-mentioned factors, have been shown to affect the regulatory T cells activity in mice models and cell cultures [47,48]. Thus, the studies considering the effect of specific bacteria on naïve T cells differentiation [41,49] or the influence of bacterial antigens in the T helper cells (Th) activity [50] have become crucial to assess the impact of those factors in the neonate microbiota. Although, the alterations of the microbiota due to perinatal and postnatal factors appear not to be maintained after neonates acquire an adult-type microbiota, at around 3–5 years of age, these coordinated events could have long-lasting effects on how the microbial community would influence the host immune responses [51].

### 3.1. Environment and Geographical Location 

The environment is a natural source of microbes to the newborn that is shared with cohabiting individuals [52], objects, and animals. Thus, neonates with siblings may have a colonization process slightly different than children without them [53]. Regarding contact between infants and pets, a beneficial effect on the prevention of some allergies and other immunological diseases has been observed [54,55]; however, the molecular mechanisms behind this effect and the potential influence of the microbiota are not clear. It has been hypothesized that rural areas, when compared with more urban zones, could have an impact on the microbiota composition and its response to an immunological challenge [56]. For instance, children living on farms have higher bacterial diversity [57] and decreased risk of developing allergies [56,58]. Notably, most microbiota studies during infancy, up to date, have been conducted in European and American populations, and information about bacterial colonization processes in tribal communities and developing countries is still scarce. In a study, De Filippo et al. observed that children living in rural African areas have higher intestinal bacterial richness and diversity compared with European children, and several exclusive short-chain fatty acids (SCFAs)-producing bacteria, whose protective role against gut inflammation has been proven [59]. Indeed, not only postnatal exposure, but also maternal exposure to this type of environment could influence the risk of the development of allergic manifestations in offspring [8]. Therefore, it is of great interest to understand how the microbiota may affect the epigenetics and modify the immune system maturation during the prenatal period. It is crucial to consider the effect of factors, such as geographical location, diet, and lifestyle upon studying the microbiota composition, because the effect observed by a specific factor could vary depending on the country where the study has been performed.

### 3.2. Mode of Delivery

The microbiota of vaginally delivered newborns resemble the maternal vaginal microbiota, which is dominated by *Lactobacillus* and *Prevotella* species, which are likely transferred to the infant through the birth canal [43]. Other microorganisms from the maternal gut microbiota, including members from the Enterobacteriaceae family, such as *Escherichia* or *Klebsiella*, have also been shown to contribute to the seeding process during vaginal birth. In contrast, neonates delivered by C-section appear to contain microbiota similar to the maternal skin, including *Staphylococcus*, *Corynebacterium*, and *Propionibacterium* [43], as well as environmental microbes [60].

Longitudinal studies comparing the influence of vaginal and C-section deliveries on an infant’s gut microbiota have provided conflicting results. This may partly be due to the high variability of the infant microbiota after birth and to differences in methodological procedures. Consistently, it has been reported that vaginally born infants have a higher abundance of the *Bifidobacterium* genus compared with those born by C-section, as observed during the first week of life [42,44,61]. Some studies suggest that the differences are still maintained at three months of life [43], although other studies have not confirmed these results [42]. Furthermore, neonates born by C-section have a delayed colonization of the Bacteroidetes phylum, and specifically of the *Bacteroides* genera, during the first month of life [41,42,43,44,62]. Other genera, specially *Clostridium*, have been associated with C-section deliveries [42], although this result is less consistent among studies. It has also been speculated that this difference could be caused by the antibiotics used in C-sections before and during the surgery and by the difficulty of breastfeeding during the first hour of life of those infants [63].

Little is known about the effects of microbiota alteration, considering the delivery mode, in the immune system development of the neonate. Lower levels of Th1-associated cytokines in babies born by C-section have been observed, and this could likely reflect delayed Th1 response maturation in these children, thus having an impact on the future capacity of immune system responses to environmental challenges. It has also been reported that vaginally born infants harbor higher levels of immunoglobulin (Ig)A, IgG, and IgM throughout the first year of life compared with those observed in C-section infants [35].

Lower bacterial diversity and altered microbial composition, as observed in C-section-delivered infants, may influence the immune system maturation process and play a role in diseases related to an immune imbalance, including asthma [64,65] and atopic and allergic disorders [63]. However, this relationship has been reported in observational and epidemiological cohort studies, and it is not possible to assign a causal relation between the microbiota associated with the mode of delivery and immune disorders.

Interestingly, animal studies have reported that vaginal delivery is associated with an early acquisition of lipopolysaccharide (LPS) resistance, which is mediated by activation of epithelial cells that may induce systemic immune tolerance [66]. In humans, less LPS biosynthesis was observed to have a protective effect against asthma [67,68]. Moreover, the *Bacteroides* genus, and specifically *B. fragilis*, could be a source of LPS and may therefore have the ability to activate regulatory T cells upon toll-like receptor 2 (TLR2) activation by LPS surface polysaccharide A (PSA) [69]. However, not all *Bacteroides* species have the same LPS types and not all of them may trigger the same immune responses. Therefore, further mechanistic studies are needed in order to evaluate the impact of the microbiota shift associated with C-sections in the molecular pathways of the immune system that can influence children’s health later in life.

### 3.3. Antibiotics Intake during Pregnancy and Early Infancy

Antibiotics intake, along with feeding habits, is the postnatal factor that primarily influences infant microbiota establishment. Intrapartum antibiotics prophylaxis, which is associated with C-sections, but also with some vaginal deliveries of group B streptococcus-positive mothers, has been observed to influence the microbial colonization process of a newborn [70,71]. Similarly, antibiotics consumption during pregnancy may affect the gestational process and the maternal and neonatal microbiota composition [72,73].

Higher levels of Proteobacteria with a detriment in Actinobacteria species and decreased microbial diversity have been associated with antibiotic use during the first month of life [45,46]. An effect of antibiotic intake on the risk of obesity [55,74,75,76], asthma and other atopic diseases [77,78,79], diabetes [80,81,82], and even neurocognitive outcomes [83] has been proposed. However, the impact of long-lasting antibiotics consumption in infants is poorly understood partly due to the low frequency of those long treatments in children. On the contrary, studies on animal models have reported reductions of the *Lactobacillus* genera and segmented filamentous bacteria after antibiotic administration, resulting in the T helper 17 cells activation in the colon [84,85]. Antibiotics also affected diversity, motility, and toll-like receptor (TLR) expression in mice colons [86]. Furthermore, antibiotics treatment can modify the homeostasis between bacterial, fungal, and viral microbiota components (50). For example, antibiotics can cause the overgrowth of fungal species, particularly *Candida albicans*, with the induction of mast cells and inflammatory interleukin (IL) expression, such as IL-5 or IL-13 [87].

## 4. Postnatal Microbial Establishment and Immune Maturation

Gut bacteria represents the key player in microbial stimulation, providing specific signals for adequate immune stimulation and development [89]. Infant microbial colonization develops during a critical time window, which includes the first thousand days of life [90], and that will favor the mutualistic relationship between the host and microbiota, contributing to immunological homeostasis. Early and diverse microbial establishment with repeated exposure to foreign microbial antigens (including polysaccharide A and LPS) and short-chain fatty acids would be key inthe specific microbial species colonization and its influence on driving normal immune mucosal and systemic maturation [89,91]. It is believed that the *pioneer microbiota*, consisting of Firmicutes, Bacteroidetes, Enterobacteriaceae, *Veillonella*, and especially *Bifidobacterium* [91,92,93,94], is responsible for the initial education of the evolving immune system, as it provides a favorable environment for further microbial settlement, including the production of an anaerobic milieu, specific compounds, and protection from the systemic immune system [92]. The early intestinal microbiota is characterized by low diversity and a relative dominance of the phyla Proteobacteria and Actinobacteria. Subsequently, the microbiota becomes more diverse, thus resembling an adult composition by 2–5 years of age [18]. However, the microbial community development and species interactions seem to be influenced by the order and timing of species immigration to the host tissue [51]. For instance, changes in the gut microbiota during the first months of life, such as shifting from facultative anaerobes to strict lactic acid-producing anaerobes, might partially be a consequence of the infant’s history exposure and the patterns of colonization from their mother [62,95]. These events suggest that a priority effect of microbial colonization may have long-lasting consequences during the early stages of the gut microbiota development, although the importance for the host health remains unexplored [51].

The settlement of bacteria on the surface of the gut can protect against pathogen penetration in a process known as ‘colonization resistance’, which is of great importance for the prevention of pathogen-induced gastrointestinal inflammation [96]. Commensal bacteria have been shown to control potential pathogen infections by competition for nutrients, adhesion sites, pH, receptors, and production of specific metabolites and antimicrobial peptides [97] by creating a hostile environment for pathogen survival and establishment (for instance, lactobacilli is able to reduce local pH and produce some anti-pathogenic compounds in the vagina).

### Early Colonizers and the Immune System

It is likely that microbial colonization of the fetal gut might direct initial maturation, playing an essential role for a balanced postnatal innate and adaptive immune system (Figure 1). However, it is still not fully understood how the neonatal gut tissue adapts to continuous microbial exposure, but features of the maternal breast milk (BM) are considered to direct some of these early responses to commensal microbes [93]. Also, the neonatal immature immune system during the first period of life might also explain its ability to accept and tolerate the microbiota establishment. The early and developing immune system is characterized by dampened pro-inflammatory cytokine responses, resulting in a more regulatory profile [98], which favors the establishment of the microbiota. It has been reported that early neonatal gut exposure to commensal bacteria can also suppress cells inducing pro-inflammatory responses, such as invariant natural killer T (iNKT) cells, thus limiting mucosal inflammation [99].

The microbial pleiotropic effect on the host includes its involvement in shaping the architecture of the immune system (including Peyer’s patches), in regulating the balance between the immune cell types present and in influencing the development of immune cell populations (including regulatory T cells) [93]. Neonatal immune cells vary in function from adult ones mainly in their tolerance in responses upon antigen exposure [91]. The two main gastrointestinal immune compartments that are affected by the colonizing microbiota are the mucus layer and the gut-associated lymphoid tissue [93]. The mucosal barrier here is constantly in contact with the enteric inhabitants, and it is responsible for providing a powerful and efficient protection against pathogens while at the same time sustaining the tolerance against the normal commensals [99].

## 5. Commensal Recognition and Tolerogenic Features of the Gut Immune System

### 5.1. Intestinal Epithelial Cells

Intestinal epithelial cells (IEC) provide both a physical and chemical barrier, thus maintaining intestinal homeostasis [100]. These cells produce various immunoregulatory signals, as a result of host–commensal microorganism interactions, which are essential for tolerant immune cells of the gut, limiting the steady-state inflammation, as well as directing appropriate immune responses against pathogens and commensal bacteria [101,102]. IEC-derived IL-25 cytokine, thymic stromal lymphopoietin, or retinoic acid together with transforming growth factor-β are important for the priming of adaptive immune cell responses and homeostasis, as well as for the regulation of innate effector responses [101].

### 5.2. Microbial Associated Molecular Patterns and Dendritic Cells

Microbial-associated molecular patterns (MAMPs) are essential for maintaining the dialogue between the host and the microbiota, thus promoting healthy microbial colonization [95]. Even though neonate innate cells express TLRs (that upon stimulation in adults normally result in the production of inflammatory mediators), their responses to microbial ligands are dominated by regulatory cytokines, including IL-10 [103]. TLR signaling also elicits tolerogenic phenotypes of dendritic cells (DCs), resulting in an immune response dominated by regulatory T cells [104]. DCs possess a C-type lectin receptor that recognizes and binds to MAMPs on the microbiota, including commensal bacteria *Lactobacillus reuteri* and *Lactobacillus casei*, thus resulting in Treg cell induction [105,106]. Moreover, *Bifidobacterium infantis* stimulates DCs to induce Foxp3regulatory T cellsand IL-10-secreting T cells, and likely, each bacterium is differently sensed by DCs according to the type of pattern recognition receptors (PRRs) with which it interacts [107,108]. Furthermore, activated DCs may also induce Th1 responses and secrete anti-inflammatory cytokines, thus promoting intestinal mucosal homeostasis [109,110]. Intriguingly, even though both commensals and pathogens possess MAMPs, the commensal bacteria, including *Lactobacillus* spp. and *Bacteroides* spp., are capable of inhibiting the Nuclear Factor Kappa Beta (NF-κB) pathway, which is important in microbial recognition and subsequent inflammatory responses [95]. Similarly, the microbiota, specifically segmented filamentous bacteria, regulates the upregulation of Th17 inflammatory cells, thus protecting against infections at the mucosal surface [111].

### 5.3. IgA Antibody–Microbiota Interactions

A major defensive mechanism on the mucosal surface is comprised of secretory immunoglobulin A (sIgA), which is involved in blocking the adhesion of microbes to epithelial cells without starting any inflammatory reaction that can cause tissue damage [112]. Mucosal IgA represents the antibodies with high affinity, neutralizing microbial toxins and invasive pathogens, and low affinity, confining commensals in the intestinal lumen [112,113]. The involvement of the gut microbiota in instructing sIgA is believed to occur in two ways—either by controlling the production of sIgA [114] or by contributing to the modulation of the sIgA repertoire [112,115,116]. The production of IgA is assumed to be modified by the uptake of luminal commensal bacteria (or their antigens) by M cells (microfold cells incorporated among epithelial cells of the mucosal barrier) and in turn captured by DCs in Peyer’s patches. This results in the activation and maturation of B cells in lymph nodes leading to IgA production and thus to prevention of microbiota penetration through the mucosal barrier, a mechanism known as a negative feedback-loop [117]. Thus, the microbiota has an impact on B cells’ capacity to drive isotype switching [118], where for instance, mice microbiota alterations due to large spectrum antibiotics or germ-free conditions can lead to a switch to IgE rather than IgA and subsequent activation of basophils, mast cells. and inflammation [119]. Moreover, selected commensals can induce systemic IgA-mediated immunity, thus constitutively protecting against bacterial sepsis in mice models [120]. Overall, IgA might be important for diversity and balance of commensals by controlling their expansion in the gut, which is evident in their role in gene-expression regulation of commensals, thus selecting species that possess less inflammatory activity on the host’s tissue [121]. Intriguingly, it has been showed that intestinal IgAs are polyreactive and coat a broad but defined subset of the microbiota [122]. This suggests that the IgA, though originating from adaptive immunity, have innate-like recognition characteristics that may assist the adaptation to a broad range of the microbiota and their antigens/metabolites encountered at the mucosal surface of the gut [122].

Interestingly, the pattern of bacterial recognition by IgA in infant feces has been shown to be aberrant during the first months of life in children that develop allergies or asthma years later. From an applied point of view, this suggests that the IgA coating pattern with bacteria can be used as an early diagnostic marker of allergy risk [123].

### 5.4. Regulatory T Cells 

Although immunological tolerance in the gut is likely achieved and sustained via multiple and redundant mechanisms [124], Foxp3 regulatory T (Treg) cells have a central role in our understanding of this process. The evidence suggests that gut commensals, including *Lactobacillus*, *Bifidobacterium*, *Bacteroides*, *Clostridium*, and *Streptococcus* [125], as well as bacterial metabolites, such as butyric acid and propionic acid [48], may program Treg cells in the intestine [126,127]. For instance, *Clostridium cluster XIVa* and *Clostridium cluster IV*, have been shown to affect the number and function of Foxp3+CD4+Treg cells [128,129]. Also, *Lactobacillus rhamnosus GG* and/or *Bifidobacterium infantis* were observed to endorse the expansion of CD4+CD25+Foxp3+Treg in the gut of mice models [130], while *B. fragilis* is capable of increasing the suppressive capacity of Tregs by downregulating pro-inflammatory cytokines IL-8 and interferon gamma (IFNγ) [131]. The induction of Treg cells is proposed as one of the mechanisms of action of health-associated bacteria recognized to induce a health benefit to the host. However, certain helminths directly drive Treg cell responses in the host, thus suppressing the collateral damage during parasite infection and limiting allergic, autoimmune, and inflammatory reactions [132]. On the other hand, helminth infections may reduce vaccine responses and increase susceptibility to coinfections with other parasites [132].

The importance of the “healthy” gut microbiota is understood through the germ-free mice studies where it has been revealed that the microbiota plays a critical role in lymphoid structure development. For instance, this effect is mainly evident in the smaller size of Peyer’s patch and also a reduced amount of CD4^+^ and CD8^+^ cells and IgA producing plasma cells [133,134]. Colonization with *Bacteroides* (as *B. fragilis*), *Clostridium*, *Lactobacillus* and *Bifidobacterium* group improves various defective immune responses observed in germ-free mice and upregulates the expression of genes involved in intestinal development, transport and immune protective functions [133,135,136]. Gut commensals can also contribute to the strengthening of the intestinal barrier by favoring the epithelial cell maturation and angiogenesis, as described in mice models [135,137]. In germ-free mice, the mucus barrier thickness, compactness, and mucin content are reduced, and the number and function of goblet cells decline [138].

## 6. Role of Breast Milk on Microbiota and Immunity Maturation

The most relevant post-natal factor that supports an adequate microbial colonization and drives the immune system maturation is breastfeeding [139,140,141]. Compared with formula feeding, breastfeeding has been associated with decreased morbidity and mortality in infants, and to lower incidence of gastrointestinal infections and inflammatory, respiratory, and allergic diseases [141,142,143,144].

Breast milk is considered the gold standard nourishment for the infant, and beyond that, it also contains a wide variety of bioactive compounds that dynamically change their composition over time to satisfy the needs of the growing infant [145]. After delivery, when the infant host defenses are vulnerable, BM provides protection through transference of antimicrobial and anti-inflammatory compounds, while also stimulating the maturation of the immune system [139,140,141,142,143,144,145,146]. In addition, BM is known to contain prebiotic compounds, as well as its own microbiota, both supplying the initial gut colonizers and supporting the microbial succession in the infant gut [28,147,148]. Selected immune components and microorganisms present in BM are described in Figure 2.

### 6.1. Breast Milk Microbiota

*Staphylococcus*, *Streptococcus*, and *Propionibacterium* are universally predominant in BM [149], and typical probiotic genera, such as *Bifidobacterium* and *Lactobacillus* [150,151,152,153], are commonly detected and isolated from BM. Other lactic acid bacteria, such as *Lactococcus*, *Weisella* and *Enterococcus*, as well as typical oral inhabitants, such as *Veillonella* and *Prevotella*, and skin bacteria, such as *Propionibacterium*, *Corynebacterium*, and so on, are frequently detected in BM samples [154,155,156].

In addition, shared species between maternal feces, BM, and infant feces have been identified [157]. Although information about the function of BM bacteria is scarce, several roles have been associated with them, including seeding colonizers to the infant microbiota, facilitating infant digestion, offering protection by competing with pathogens, and improving intestinal barrier functions by enhancing mucine production and reducing intestinal permeability [141]. Breast milk microbiota and other milk molecular components likely help to educate the infant’s immune system. It promotes an adequate intestinal immune homeostasis that initially influences a shift from an intrauterine Th2 predominant to a Th1/Th2 balanced response and a stimulation of T-regulatory cells by BM-specific microorganisms [158]. Furthermore, some strains isolated from BM have shown an ability to modulate both natural and acquired immunity [152,153]. Recently, several yeasts and other fungi have been detected in BM samples from healthy mothers, including *Malassezia*, *Candida*, *Saccharomyces*, and *Rhodotorula*, among others. This finding suggests that BM could be not only participating in shaping the infant microbiome, but also the infant mycobiome [159]. Future research is needed to fully understand the role of BM microbiota in the infant, but the increasing evidence reinforces its importance on infants’ protection and training of the immune system during the first months of life.

### 6.2. Human Milk Oligosaccharides

One of the major protective roles of BM resides on human milk oligosaccharides (HMOs), which are complex carbohydrates present in high concentrations in milk (5–20 g/L) [160]. Human milk oligosaccharides are responsible for promoting the growth of beneficial commensal bacteria, such as *Bifidobacterium* and *Lactobacillus*, which is reflected in differences observed in the intestinal microbiota of breast-fed and formula-fed infants [148,161]. Human milk oligosaccharides are essentially indigestible by the human gut and reach the colon where they serve as substrate for fermentation to bacteria, such as bifidobacteria and lactobacilli and also *Bacteroides* and *Staphylococcus*, playing an important role in shaping the infant microbiome [150,151]. This fermentation results in sub-products, such as lactate and Short-chain fatty acids (SCFAs), including acetate and butyrate, and other metabolites. SCFAs also represent the main energy source for colonocytes and provide important immune protective functions. For example, acetate stimulates the proliferation of Treg cells in the lamina propria, regulating intestinal homeostasis [162], and several immunological functions for butyrate have been reported, many of them associated with potent regulatory effects on gene expression [163]. Importantly, specific HMOs can act as homologous to infant gut cell receptors, inhibiting the binding of pathogens to the intestinal epithelium and therefore offering protection against infections [160,164].

### 6.3. Secretory IgA 

Secretory IgA (sIgA) is the most abundant immunoglobulin in BM, with highest levels in colostrum (12 g/L) that decrease in mature milk (0.5 g/L). It represents the key anti-infective component in BM as it protects mucosal sites by interfering with the adherence of pathogens to epithelial cells and neutralizes toxins [165,166]. Secretory IgA transference to the infant’s gut through breastfeeding compensates the delay of a newborn’s innate sIgA production, and its concentration in milk decreases as the infant’s endogenous levels increase [167]. When an enteric pathogen is presented to a dendritic cell in the maternal intestine, activated T lymphocytes stimulate B lymphocytes, which migrate to the mammary gland where they differentiate into plasma cells and produce large amounts of IgA. IgA attaches to the polymeric Ig receptor and is transported through the mammary epithelial cell, after which the complex is cleaved and sIgA is secreted to the milk [166]. In the infant gut, sIgA provides the infant with a unique immunological protection against pathogens, including bacteria, fungi, viruses, and parasites, to which the maternal immune system has been exposed [148,168,169,170]. Levels of sIgA in milk from mothers delivering preterm are higher, showing an adaptation of BM to increase the protection of premature infants, whose immune systems are more susceptible to infections [171]. Recently, altered sIgA responses towards the gut microbiota were observed in exclusively breastfed children who later in life developed allergies, meaning that divergent antibodies/microbiota transmitted through breast milk could affect the infant’s correct immune development [123].

### 6.4. Anti-Microbial Bioactive Proteins

Breast milk contains typical components of the innate immunity that are lacking in an infant’s immature host defenses, which protect the infant while also dictating additional selection on the infant gut microbiota. Among the broad diversity of bioactive proteins, lactoferrin, lysozyme, and α-Lactalbumin are, together with immunoglobulins, major whey proteins that exert immunity-enhancing properties. Lactoferrin is present in BM in high concentrations and protects the infant against pathogens through diverse modes of action. Lactoferrin chelates free iron, a necessary nutritional requirement for most bacterial pathogens, thus inhibiting the growth of a broad spectrum of bacteria [172]. In addition, partial digestion of lactoferrin leads to lactoferricin, a peptide with potent broad activity against bacteria, fungi, and viruses [173,174,175]. Other beneficial actions of lactoferrin include the stimulation of pathogen phagocytosis by macrophages [176], the downregulation of inflammatory cytokine production in monocytic cells by its interference with NF-κB protein complex activation [177], the promotion of beneficial gut microbiota in the infant [178], and so on. Lysozyme, although present in lower concentrations than lactoferrin, is a crucial factor of the protective activity of BM. It breaks peptidoglycans bonds of bacterial cell walls, thus lysing pathogenic Gram-positive bacteria, while it can also act synergistically with lactoferrin to kill Gram-negative bacteria [179]. α-Lactalbumin is the major whey protein in BM, and some of its proteolytic fragments have shown prebiotic properties in vitro by stimulating the growth of beneficial bifidobacteria [180].

In addition, caseins are a family of highly glycosylated proteins that constitute approximately 40% of total BM proteins. κ-Casein, a minor casein subunit in BM has been shown to act as a binding analogue of *Helicobacter pylori*, inhibiting its binding to human gastric mucosa in vitro [181].

### 6.5. Cytokines

Breast milk contains several cytokines at physiologically relevant concentrations, including Interleukin (IL)-1β, IL-6, IL-8, and IL-10, especially in colostrum, which regulate and modulate the immune system and inflammatory responses [182]. Interleukin 10, a key immunoregulatory and anti-inflammatory cytokine, has been shown to inhibit blood lymphocyte proliferation [182]. Mutations in the encoding genes of the IL-10 receptor have been observed in infants developing early-onset colitis, resulting in hyperinflammation of the intestine [183]. Similarly, mice with a disrupted IL-10 gene spontaneously developed enterocolitis after weaning, and the condition was associated with unpaired cytokine production by CD4^+^ Th1-like T cells, as well as activated macrophages [184]. Intestinal inflammation could be prevented by parenteral administration of the IL-10, thus suggesting a crucial role of BM IL-10 on the infant intestinal homeostasis and prevention of exacerbated response to foreign antigens. Transforming growth factor (TGF-β) is associated with the regulation of T-cell activation, but it also affects B cells, natural killer (NK) cells, macrophages, and dendritic cells [180]. TGF-β can reduce inflammation by decreasing pro-inflammatory cytokines production while favoring IgA production, thus increasing intestinal immunity [180]. Infants fed with BM containing high concentrations of TGF-β, have been associated with lower risk of developing atopic diseases and wheezing [185,186].

### 6.6. Cells and Other Immunological Compounds

**Leukocytes.** Activated live leukocytes are present in BM, including macrophages (40–50% of total leukocytes), polymorphonuclear neutrophils (40–50% of total leukocytes), and lymphocytes (5–10% of total leukocytes) [187]. Leukocytes concentrations are the highest in colostrum and decrease gradually. If the mother or infant suffers an infection, leukocytes levels can increase to 97% of the total cells in BM [188]. The lymphocytes population consists mainly of T cells (approximately 80%), where most of them are activated, motile, and interactive, thus suggesting that they confer active immunity to the infant. In addition, leukocytes, particularly polymorphonuclear, offer protection to the mammary gland under mastitis infections [189].

**Soluble CD14 (sCD14).** sCD14 is a pattern recognition receptor, which together with toll-like receptor 4 (TLR4) and LPS-binding protein forms the lipopolysaccharide (LPS)-recognition complex that binds bacterial LPS. It participates in the signal transduction initiation of TLR-4, which results in a high secretion of IL-12, a pro-inflammatory cytokine. sCD14 has been suggested to be an important immune modulator for intestinal homeostasis and may play an important role in the infant’s intestinal defense [190]. Moreover, this protein has been associated with lower risk for asthma in early childhood [191].

**Glycosylated proteins.** Certain glycosylated proteins, such as strongly glycosylated BM mucins, can interfere with bacterial and viral adherence to the gut epithelium. Mucin-1 (MUC1) is the predominant mucin in BM, present in the surface of fat globules, and inhibits the binding of pathogens to the mucosal epithelia [191], including pathogenic bacteria, such as *H. pylori* and rotavirus, which represent the main cause of acute gastrointestinal infectious in infants [192].

## 7. Concluding Remarks

The maturation of the immune system is accomplished through bidirectional interactions where the gut microbiota directs the development of the immune system, while the host immune system shapes the establishment of the gut microbiota. Even though current studies are attempting to associate specific microbes to unique immunological states, it is worth remembering that the microbiome is a highly dynamic, diversified, and complex human organ, where all the composing microbes are expressing a vast number of potential ligands and metabolites. Under normal homeostatic conditions, both inflammatory and regulatory features are highly balanced, thus leading to the establishment of stable tissue immunity and limited inflammation. Hence, upon alterations within the microbiota that may cause an inflammatory state, it is unlikely that the outcome observed is due to a single microbial product, but rather a result from a shifted balance state of an entire community.

Several prenatal and perinatal factors, including the mode of delivery, antibiotics consumption, diet, and other environmental factors, may influence the microbial colonization of the infant and in turn its immune system maturation. Here, breast milk plays an essential role, guiding a normal microbiome development in the infant through transference of prebiotic compounds that support the colonization of beneficial bacteria and antimicrobial components that protect the infant against infections. Breast milk microbiota should also be considered as an important source of microorganisms to the infant microbiomes and as part of microbial tolerance and immune training in the child. In addition, BM contains several bioactive and immunomodulatory compounds, which together with immune cells and microorganisms support the infant’s protection when its immune system is still immature (Figure 3). Most of the existing literature is focused on the role of bacteria, but more studies need to elucidate the importance of archaea, fungi, and non-pathogenic viruses that also may be important for early gut immunity maturation. Finally, the molecular mechanisms involved in the interactions between early life microbiota and the immune system are still to be clarified.

## Figures and Tables

**Figure 1 medsci-06-00056-f001:**
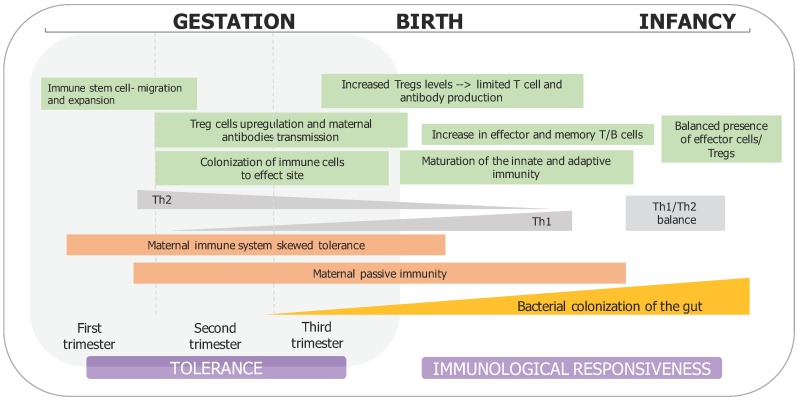
Major events occurring in immune system and gut development. During pregnancy, the T helper cells of the maternal immune system are skewed towards T helper (Th)2-type immunity, to maintain the tolerance of the developing fetus. The mother here contributes with tolerogenic mediators through the placenta (including antibodies and growth factors), thus instructing fetal immune system development. However, during the first weeks and months of life, infants subsequently increase Th1 activity, thus restoring the balance of helper T cells. Without this shift, Th2 persistence might be associated with atopic diseases including asthma. While Th cells play an important key role in the direction of immune responses, mainly neonatally, regulatory T cells suppress the activation and development of naïve T cells towards Th types, thus maintaining the mucosal homeostasis both during the pregnancy and the infancy period [88].

**Figure 2 medsci-06-00056-f002:**
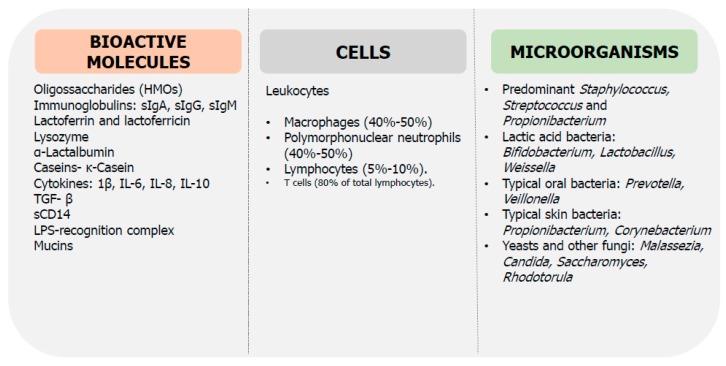
Selected microbial and immunological factors in breast milk (BM). sIg: secretory immunoglobulin, IL: interleukin, TGF: transforming growth factor, LPS: lipopolysaccharide.

**Figure 3 medsci-06-00056-f003:**
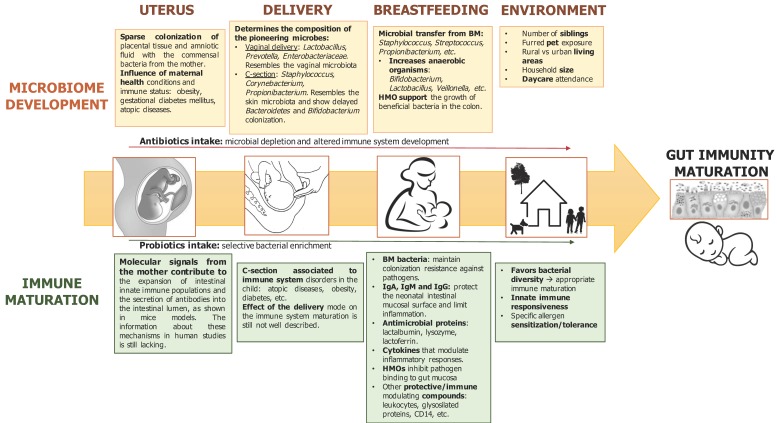
Key factors influencing microbiota development and maturation of the immune system in early life. BM: breast milk; and HMO: human milk oligosaccharides.

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
