# Peer review of "Gut Microbiota and Mucosal Immunity in the Neonate"

_medsci, 2018, doi:10.3390/medsci6030056_

Round 1

Reviewer 1 Report

The topic of this review “Gut microbiota and mucosal immunity in the neonate” is timely and important.  The authors have summarized the literature concerning the influence of the microbiota in the neonate.

Specific points:

-       Based in their own publications together with some other studies the authors took position and assumed the presence of life microorganisms in contact with the fetus and colonizing the different mucosal sites. The authors consider these microorganisms as a commensal microbiota and distinguee them from the maternal derived infectious microbes. The authors only lightly announced the controversy of these results due to the low frequency and inconsistent results (sequenced (30) versus culturable bacteria (16,19)).  Therefore, without relevant scientific evidences, the authors make strong assumptions all along the text such as “… 140 Nevertheless, it is still not fully clear if the microbes that colonize the fetus may be of beneficial feature for pregnancy outcome 141 and the future health status of the neonate….” “…Likely, microbial settlement of the fetal gut might 280 direct initial maturation, playing an essential role for a balanced postnatal innate and adaptive 281 immune system…”.  This point should be deep discussed in the text.

-       Contrarily, in the Figure 2, to summarize the impact of the microbiota on the prenatal immune maturation, the authors used the results obtained from a mice study demonstrating that only inert maternal bacterial derived products and not life bacteria are transferred from the mother gut to the fetus (14). Using these results in the Figure 2, the authors could induce the reader to the wrong conclusion.

-       The authors describe the microbiota in the meconium as “…contains a complex microbial community…”. The authors said that the detected microorganisms in the meconium appear to be present in low frequency. Moreover, the metabolic activity of the detected microorganisms in the meconium has not been studied. Therefore, what do the authors mean by complex?

-       The quality of the English syntax used by authors should be improved. Repeatedly, the sentences are too long (some over five lines), identical words are used in the same sentences, some expressions and concepts appear redundantly in the text, and 3 to 4 conjunctions are used in the same sentence. The reader would struggle with it and lose the message of the review. 

-     The authors use microbiota and microbiome as synonymous what is clearly not the case. These concepts should be described, eventually in a box, and used correctly in the text.

-       There is an issue with the reference call. The reference 111 is called between the 35 and 38.  Some references are not called in the text, as 36, 37.

-       The authors preferentially cited articles published earlier than 4 years ago (aprox. 65 %). Nevertheless, a lot of the research about the host-microbial relationship and in particularly during the neonatal period have been done in this period. Recent references in the field of “critical window and later on health consequences” are missing and thus, not included in the discussion.

Author Response

REVIEWER 1

 Comments and Suggestions for Authors

The topic of this review “Gut microbiota and mucosal immunity in the neonate” is timely and important.  The authors have summarized the literature concerning the influence of the microbiota in the neonate.

Authors answer:Thank you for the comment. We are very pleased that the reviewer found our work of interest and we have greatly appreciated the time and expertise dedicated by the reviewer to the constructive criticism of this article. We have made an extensive revision of the manuscript, taking into account all of the reviewers’ suggestions and comments, which have helped us to clarify interpretative issues and limitations, and significantly improve the manuscript.

 Specific points: 

-       Based in their own publications together with some other studies the authors took position and assumed the presence of life microorganisms in contact with the fetus and colonizing the different mucosal sites. The authors consider these microorganisms as a commensal microbiota and distinguee them from the maternal derived infectious microbes. The authors only lightly announced the controversy of these results due to the low frequency and inconsistent results (sequenced (30) versus culturable bacteria (16,19)).  Therefore, without relevant scientific evidences, the authors make strong assumptions all along the text such as “… 140 Nevertheless, it is still not fully clear if the microbes that colonize the fetus may be of beneficial feature for pregnancy outcome 141 and the future health status of the neonate….” “…Likely, microbial settlement of the fetal gut might 280 direct initial maturation, playing an essential role for a balanced postnatal innate and adaptive 281 immune system…”.  This point should be deep discussed in the text.

Authors answer: Thank you for the comment. We agree with the reviewer comment about the in utero colonization hypothesis has not been fully explored and still remains controversial. We have changed the general orientation for this topic along the manuscript. We no longer state that the fetus could be colonized by uterine bacteria, rather that the bacterial DNA was encountered in the placenta. However, viable bacteria have been isolated from meconium, including anaerobes that unlikely are the result of environmental contamination. We have also stated the technical issues regarding the discrimination of contaminates, in next generation sequencing studies, due to low bacterial load of the samples (Please, see line 139-148)

-       Contrarily, in the Figure 2, to summarize the impact of the microbiota on the prenatal immune maturation, the authors used the results obtained from a mice study demonstrating that only inert maternal bacterial derived products and not life bacteria are transferred from the mother gut to the fetus (14). Using these results in the Figure 2, the authors could induce the reader to the wrong conclusion.

Authors answer: Thank you for the comment. Following the referee´s suggestions, we now indicate that most of the studies considering immune responses in uterus, related with the impact of maternal bacterial derived products, are performed in mice models and the information is human is scarce.

-       The authors describe the microbiota in the meconium as “…contains a complex microbial community…”. The authors said that the detected microorganisms in the meconium appear to be present in low frequency. Moreover, the metabolic activity of the detected microorganisms in the meconium has not been studied. Therefore, what do the authors mean by complex?

Authors answer: We agree with the reviewer’s comment and we cannot conclude that meconium harbors a complex bacterial community, due to the lack of studies. This has now been changed in the text (Please, see line 83-89)

-       The quality of the English syntax used by authors should be improved. Repeatedly, the sentences are too long (some over five lines), identical words are used in the same sentences, some expressions and concepts appear redundantly in the text, and 3 to 4 conjunctions are used in the same sentence. The reader would struggle with it and lose the message of the review.  

Authors answer: Thank you for the comment. We have revised the English syntax and also, we have tried to avoid long sentences. In general, we have focused the content of this review.

-     The authors use microbiota and microbiome as synonymous what is clearly not the case. These concepts should be described, eventually in a box, and used correctly in the text.

Authors answer: Thank you for the comment. We have revised the manuscript and we have homogenized the terms. We have included a box with the definitions and we have used both terms , “microbiota” and “microbiome”, properly.

-       There is an issue with the reference call. The reference 111 is called between the 35 and 38.  Some references are not called in the text, as 36, 37.

Authors answer: We have checked the references and we have solved the issue highlighted by the reviewer.

-       The authors preferentially cited articles published earlier than 4 years ago (aprox. 65 %). Nevertheless, a lot of the research about the host-microbial relationship and in particularly during the neonatal period have been done in this period. Recent references in the field of “critical window and later on health consequences” are missing and thus, not included in the discussion.

Authors answer: Thank you very much for the comment. We have revised and updated the references included. More recent literature is now mentioned in the text.  

Reviewer 2 Report

Gut Microbiota and mucosal immunity in the neonate

Dzidic, Boix Amoros et al

This manuscript very elegantly describes the microbiota of the developing infant during gestation and after birth, and the factors that contribute to the microbiota and simultaneous immune system development. The manuscript is well presented and logical, my only suggestion would be to include, if possible, some information on the infant microbiome and immune system development of tribal communities.

The abstract would benefit from some editorial revision but the rest of the manuscript reads very well and there are only a few minor changes that I would recommend.

I suggest the following:

Line 32: start sentence with “Early microbial exposure starts during gestation…..”

Line 36  replace “evolution” with “development”

Line 37 replace “evolves” with” happens”

Line 39 “summarizes” rather than “summarized”

Line 39 change to “evidence to date on early gut……”

Line 41 antibiotic rgather than antibiotics

line 54 change “organism”  to “host”

Line 62 change “will “to “can”

Line 69 change “needed “ to “required”

Line 111 change to “ The bacteria composition of meconium includes Enterococcus and Escherichia, and partly resembles….”

Line 251 change “evolves” to “develops”

Line 411 change ”seem to be “to “are”

Line 414 remove the “o”

Throughout the manuscript please change “host-microbes interactions” to” host-microbe interactions”

Author Response

REVIEWER 2

 Comments and Suggestions for Authors

 This manuscript very elegantly describes the microbiota of the developing infant during gestation and after birth, and the factors that contribute to the microbiota and simultaneous immune system development. The manuscript is well presented and logical, my only suggestion would be to include, if possible, some information on the infant microbiome and immune system development of tribal communities.

 Authors answer: Thank you for the comments. We agree with the reviewer’s comment that most of the existing studies are conducted in Europe and US and therefore, we have now added a paragraph describing this topic. (Please, see line 175-181)

The abstract would benefit from some editorial revision but the rest of the manuscript reads very well and there are only a few minor changes that I would recommend. 

Authors answer: We have now extensively revised the abstract.

 I suggest the following:

 Line 32: start sentence with “Early microbial exposure starts during gestation…..”

 Line 36  replace “evolution” with “development”

 Line 37 replace “evolves” with” happens”

 Line 39 “summarizes” rather than “summarized”

 Line 39 change to “evidence to date on early gut……”

 Line 41 antibiotic rgather than antibiotics

 line 54 change “organism”  to “host”

 Line 62 change “will “to “can”

 Line 69 change “needed “ to “required”

 Line 111 change to “ The bacteria composition of meconium includes Enterococcus and Escherichia, and partly resembles….”

 Line 251 change “evolves” to “develops”

 Line 411 change ”seem to be “to “are”

 Line 414 remove the “o”

 Throughout the manuscript please change “host-microbes interactions” to” host-microbe interactions”.

Authors answer: Thank you for the suggestions. All points and issues raised by the reviewers have been corrected and modified. We are very pleased that the reviewers found our work of interest and we have greatly appreciated the time and expertise dedicated by the reviewers to the constructive criticism of this article. We have made an extensive revision of the manuscript, taking into account all of the reviewers’ suggestions and comments, which have helped us to clarify interpretative issues and limitations, and significantly improve the manuscript.

Round 2

Reviewer 1 Report

Dzidic and colleagues have revised their manuscript substantially and addressed my concerns. 

Minor editor modifications still required prior publications: correct use for abbreviations (i.e. BM, IL, )  and some homogenization (i.e. Foxp3 and FoxP3).